# The effect of baseline serum uric acid on chronic kidney disease in normotensive, normoglycemic, and non-obese individuals: A health checkup cohort study

**Young-Bin Son, Ji Hyun Yang, Myung-Gyu Kim, Sang Kyung Jo, Won Yong Cho, Se Won Oh** *

Division of Nephrology, Department of Internal Medicine, Korea University Anam Hospital, Korea University College of Medicine, Seoul, Republic of Korea

* hisy81@hanmail.net

## Abstract

### Introduction

The independent role of serum uric acid (SUA) on kidney disease is controversial due to its association with metabolic syndrome. The objective of this study was to investigate the association of baseline SUA with development of chronic kidney disease and eGFR decline in normotensive, normoglycemic and non-obese individuals during follow up period.

### Materials and methods

We included non-hypertensive, non-diabetic, and non-obese 13,133 adults with estimated glomerular filtration rate (eGFR) $\geq$ 60ml/min/1.73m$^2$ who had a voluntary health check-up during 2004–2017.

### Results

SUA was positively related to adjusted means of systolic blood pressure (SBP), triglyceride, body mass index, and body fat percent. SUA was inversely associated with high density lipoprotein HDL (P for trend $\leq$0.001). SUA was an independent risk factor for the development of diabetes, hypertension, and obesity. During 45.0 [24.0–76.0] months of median follow up, the highest quartiles of SUA showed significant risks of 30% eGFR decline compared than the lowest quartile (RR:3.701; 95% CI: 1.504–9.108). The highest quartile had a 2.2 fold (95% CI: 1.182–4.177) increase in risk for incident chronic kidney disease (CKD).

### Conclusions

SUA is an independent risk factor for the development of diabetes, hypertension, and obesity in the healthy population. High SUA is associated with increased risk of CKD development and eGFR decline in participants with intact renal function.

**Data Availability Statement:** There are ethical restrictions on sharing a de-identified data set. The ethics committee in Korea University Anam

Hospital do not approve that the patient's data be open to the public. The data is only available to the researchers whose research proposal has been examined and approved the ethics committee of Korea University Anam Hospital. If other researchers would like to access the data, they should submit their research proposal in the homepage (http://ctc.kumc.or.kr/).

**Funding:** This study is supported by a Korea University Grant (Grant No. K2008381)

**Competing interests:** The authors have declared that no competing interests exist.

## Introduction

Uric acid has been suggested as a potentially modifiable risk factor for chronic kidney disease (CKD) [1–3]. Tubulointerstitial precipitation of uric acid can lead to renal damage [4]. Regardless of uric acid crystal, animal studies have shown that uric acid causes vascular smooth muscle cell proliferation in afferent arteriole, tubulointerstitial inflammation, insterstitial fibrosis, and eventually renal dysfunction [5].

However, recent clinical studies did not show consistent results about the effect of uric acid on renal outcomes. Some small controlled clinical studies have reported that urate lowering therapy could slow CKD progression [6–9]. In contrast, other studies did not show the beneficial effect of urate lowering therapy in patients with diabetic nephropathy, IgA nephropathy, or CKD stage 3 [10–12]. A recent large scaled clinical trial has reported that urate lowering therapy does not reduce the decline of renal function in CKD 3 patients. However, it has significant benefits for those whom serum creatinine level is lower than median or in patients without proteinuria [13].

On the other hand, increasing intake of purine and fructose diet can cause hyperuricemia and obesity [14]. Hyperuricemia is associated with insulin resistance, hypertension (HTN), and diabetes mellitus (DM) [15–18]. These metabolic factors can lead to renal damage and they are confounders when analyzing the effect of uric acid on renal outcome.

Therefore, we planned longitudinal studies for assessing development of CKD according to baseline serum uric acid (SUA) in participants who had intact renal function. We investigated the effect of SUA in participants who did not have metabolic abnormalities such as DM, HTN or obesity to assess the independent association between SUA and renal outcome.

## Materials and methods

### Participants

Participants were those who underwent general health check-up during 2004–2017 in Korea University Anam Hospital. We included 55,098 participants aged $\geq$18 years with estimated glomerular filtration rate (eGFR) $\geq$ 60 ml/min/1.73m$^2$. Among them, we exclude 38,103 participants whose serum creatinine were checked only one time and 3,562 participants who had HTN (N = 2,743), DM (N = 797) or obesity (N = 492) (Fig 1). We divided 13,133 participants into gender-specific quartile groups according to serum uric acid. This study was approved by the Institutional Review Board of Korea University Anam Hospital Clinical Trial Center (IRB No. 2019AN0181). It was conducted in accordance with the Declaration of Helsinki. Institutional review board approved that informed consent is not necessary because this is a retrospective study.

### Measurements

Blood samples after an 8-hour fast were collected year-round and immediately processed, refrigerated, and transported (in cold storage) to the laboratory for analysis within 12 hours: serum uric acid, creatinine, hemoglobin (Hb), white blood cell (WBC), aspartate aminotransferase (AST), alanine aminotransferase (ALT), alkaline phosphatase (ALP), bilirubin, protein, albumin, glucose, cholesterol, triglyceride (TG), high density lipoprotein (HDL), low density lipoprotein (LDL), C-reactive protein (CRP) were measured. Measurement of serum creatinine was performed using Toshiba Neo (Toshiba Medical System Co, Otawara, Japan) between 2004 and Nov. 2012 or Beckman Coulter AU5811, 5821 (Diamond Diagnostics, Holliston, MA, USA) between Dec. 2012 and Nov. 2018. Serum creatinine was measured with the Jaffe kinetic method (Jan. 2004—Nov. 2012, CLINIMATE Creatinine sekisui, Sekisui Medical Co. Ltd., Tokyo, Japan; Dec. 2012—Nov. 2018; Beckman Creatinine, Beckman Coulter, Inc.,

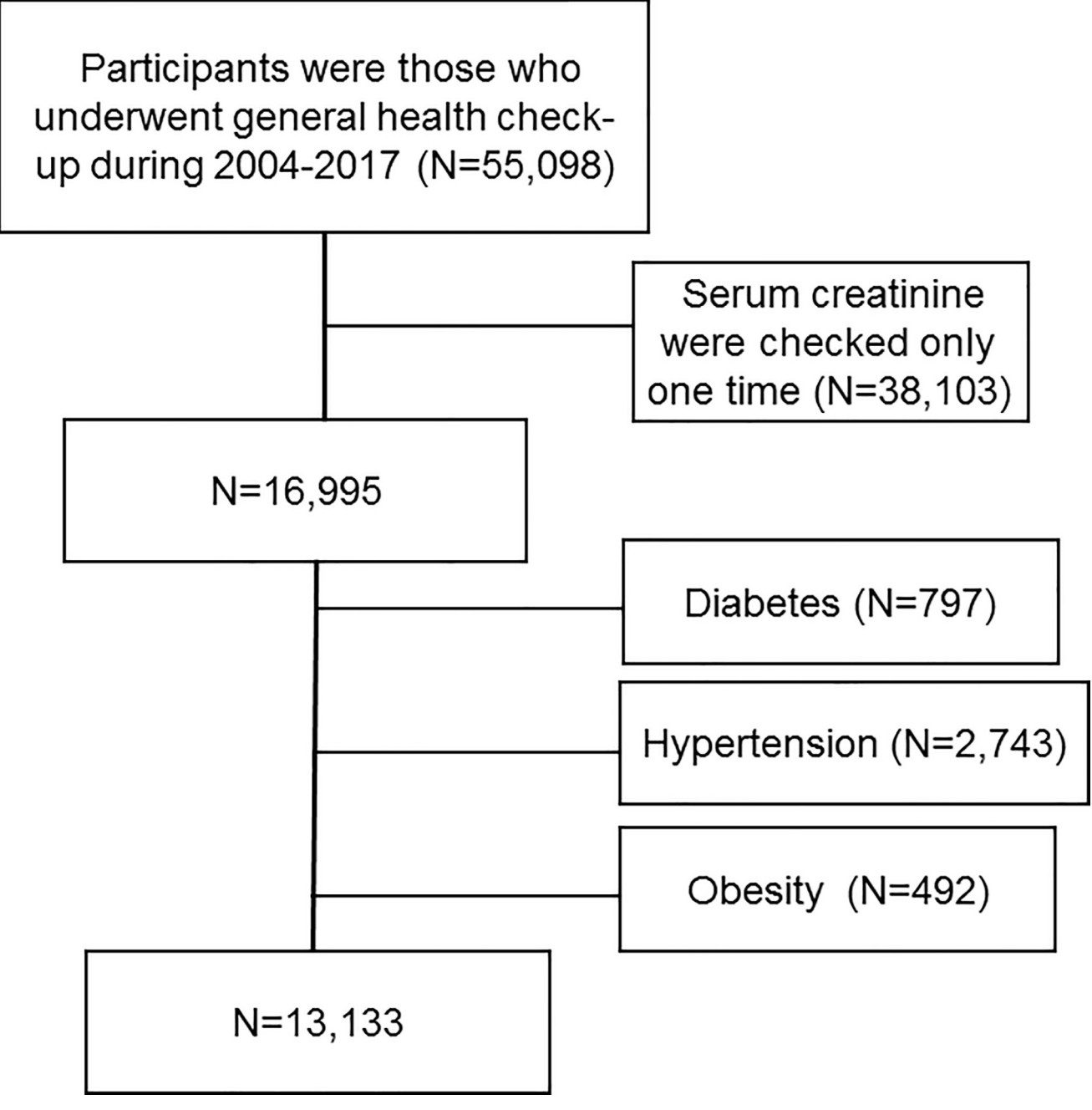

**Fig 1. Selection of study population.**

USA). GFR was estimated using the Chronic Kidney Disease Epidemiology Collaboration (CKD-EPI) equation [19]. Urine protein was measured by dipstick urinalysis. Results are reported using a semiquantitative scale from negative to 4+.

## Definitions

HTN was defined as the presence of either (i) systolic BP (SBP) ≥ 140 mmHg or diastolic BP (DBP) ≥ 90 mmHg or (ii) diagnostic code of hypertension. DM was defined as participants

who fulfilled at least one of the following three criteria: (i) fasting blood glucose (FBS) $\geq$126 mg/dL; (ii) HbA1c $\geq$ 6.5%; (iii) diagnostic code of diabetes. Body mass index (BMI) was calculated based on weight and height (kg/m$^2$). Obesity was defined as BMI $\geq$ 30 kg/m$^2$. Proteinuria was defined as dipstick urinalysis above 1+. Decline of renal function was defined as 30% of decrease of follow up eGFR compared to baseline eGFR. Incident CKD was defined as the incidence of eGFR decline below 60 ml/min/1.73m$^2$. Body fat and muscle percent were measured using bioelectrical impedance analysis (InBody770, InBody Co, Ltd., Seoul, Korea). Malignancy was defined as presence of "C" code in electronic medical record.

## Statistical analysis

All analyses were performed using SPSS software (SPSS version 25.0, Chicago, IL, USA). Data are presented as mean ± standard deviation (SD) for continuous variables and as percentage for categorical variables. Differences were analyzed using Chi-square test for categorical variables and analysis of variance (ANOVA) for continuous variables. Adjusted means and 95% confidence intervals (95% CIs) of SBP, TG, HDL, BMI, body fat percent and FBS were calculated using analysis of covariance to adjust independent factors related to SUA and post-hoc analysis was used to correct for multiple comparisons. Risks and 95% CIs of development of HTN, DM, obesity, decline in eGFR, and incident CKD were calculated using cox regression analysis. A P value < 0.05 was considered statistically significant.

## Results

### Baseline characteristics of participants

Baseline characteristics of participants are described in Table 1. The study cohort had an average age of 44.5 ± 10.2 years, an eGFR of 92.2 ± 13.6 ml/min/1.73 m$^2$, and a uric acid level of 5.2 ± 1.4 mg/dL. Males accounted for 54.5%. Higher SUA levels were significantly associated with younger age and higher levels of hemoglobin, protein, albumin, and liver enzymes such as AST, ALT, and ALP. Higher SUA was also associated with higher WBC and CRP but lower eGFR. Higher SUA was related to higher BMI, BP, FBS and dyslipidemia such as higher total cholesterol, TG, LDL, and lower HDL (Table 1).

### Association of SUA with metabolic disorders at baseline

Associations of SUA with SBP, TG, HDL, BMI, body fat percent, and FBS were evaluated after adjusting for multiple factors at baseline. SBP was positively related to SUA (P for trend = 0.001). Adjusted SBP of 4$^{th}$ quartile was significantly higher than that of the 1$^{st}$ quartile (112.1 ± 0.2 mmHg vs. 111.4 ± 0.2 mmHg, P = 0.003) (Fig 2A).

TG, BMI, and body fat percent had positive relations with SUA (P for trend < 0.001). The TG of the 3$^{rd}$ or 4$^{th}$ quartile was significantly higher than that of the 1$^{st}$ quartile (P <0.001) (Fig 2B). Adjusted means of BMI and body fat percent were higher in the 3$^{rd}$, and 4$^{th}$ quartiles than those of the lowest quartile (P $\leq$ 0.024) (Fig 2D and 2E). SUA was inversely associated with HDL (P for trend < 0.001). HDL of the 2$^{nd}$, 3$^{rd}$, and 4$^{th}$ quartiles were significantly lower than those of the lowest quartile (P < 0.01) (Fig 2C). Adjusted FBS was not associated with SUA (Fig 2F).

### Development of diabetes, hypertension, and obesity according to SUA quartiles

During 45.0 [24.0–76.0] months of follow up period, we analyzed the incidence of new onset DM, HTN, and obesity. DM, HTN, and obesity were developed in 341 (2.6%), 765 (5.8%), and

**Table 1. Baseline characteristics of study population.**

| | Uric acid | | | | |
|---|---|---|---|---|---|
| | 1st (N = 3230) | 2nd (N = 3118) | 3rd (N = 3419) | 4th (N = 3366) | |
| Age (years) | 45.5±10.0 | 44.6±10.2 | 43.8±10.0 | 44.1±10.3 | <0.001 |
| BMI (kg/m$^2$) | 22.6±2.7 | 22.9±2.6 | 23.3±2.7 | 24.0±2.7 | <0.001 |
| Men (%) | 1718 (53.2) | 1738 (55.7) | 1858 (54.3) | 1842 (54.7) | 0.223 |
| SBP (mmHg) | 111±11 | 111±11 | 112±11 | 113±11 | <0.001 |
| DBP (mmHg) | 68±9 | 68±9 | 69±9 | 70±9 | <0.001 |
| Uric acid | 3.9±0.9 | 4.9±0.9 | 5.5±1.0 | 6.5±1.2 | <0.001 |
| eGFR (mL/min/1.73 m$^2$) | 95.1±13.3 | 93.3±13.5 | 91.8±13.3 | 88.7±13.6 | <0.001 |
| Hb (g/dL) | 14.0±1.7 | 14.3±1.5 | 14.4±1.5 | 14.5±1.4 | <0.001 |
| WBC (1000/μL) | 5.6±1.6 | 5.7±1.5 | 5.8±1.5 | 6.0±1.5 | <0.001 |
| AST (IU/L) | 22.3±14.0 | 22.9±20.0 | 22.9±12.2 | 24.4±12.4 | <0.001 |
| ALT (IU/L) | 21.0±23.1 | 22.3±25.2 | 23.0±16.9 | 26.4±22.9 | <0.001 |
| ALP (IU/L) | 54.4±17.2 | 54.8±16.3 | 55.4±17.1 | 56.9±17.1 | <0.001 |
| Bilirubin | 0.81±0.36 | 0.84±0.38 | 0.84±0.38 | 0.85±0.36 | 0.001 |
| Protein | 7.2±0.4 | 7.2±0.4 | 7.2±0.4 | 7.3±0.4 | <0.001 |
| Albumin | 4.5±0.3 | 4.5±0.3 | 4.5±0.3 | 4.6±0.3 | <0.001 |
| Glucose (mg/dL) | 90.5±9.2 | 90.2±9.4 | 90.5±9.3 | 91.2±9.9 | <0.001 |
| Cholesterol(mg/dL) | 180.7±31.7 | 183.7±32.0 | 185.9±31.3 | 192.3±34.2 | <0.001 |
| TG (mg/dL) | 105.7±64.9 | 112.6±66.2 | 120.9±76.9 | 140.9±95.7 | <0.001 |
| HDL (mg/dL) | 55.7±13.5 | 54.2±12.7 | 53.8±12.8 | 52.4±13.1 | <0.001 |
| LDL (mg/dL) | 106.2±27.8 | 109.4±28.6 | 111.7±28.6 | 116.9±31.7 | <0.001 |
| CRP | 1.2±3.3 | 1.2±2.8 | 1.3±2.5 | 1.7±3.9 | <0.001 |
| Malignancy (%) | 174 (5.4) | 149 (4.8) | 184 (5.4) | 190 (5.6) | 0.462 |

Abbreviations: BMI (body mass index), systolic blood pressure (SBP), diastolic blood pressure (DBP), estimated glomerular filtration rate (eGFR), hemoglobin (Hb), white blood cell (WBC), aspartate aminotransferase (AST), alanine aminotransferase (ALT), alkaline phosphatase (ALP), triglyceride (TG), high density lipoprotein (HDL), low density lipoprotein (LDL), C-reactive protein (CRP), diabetes mellitus (DM), and hypertension (HTN).

124 (0.9%) participants, respectively. The 3rd, and 4th quartiles of SUA showed higher risk for the development of DM than the lowest quartile by multivariate analysis (RR: 1.805, 95% CI: 1.305–2.498; RR: 1.767, 95% CI: 1.279–2.442, respectively). SUA was related to the development of HTN and obesity. The 3rd and 4th quartiles had higher risk of the development of HTN (RR: 1.285, 95% CI: 1.038–1.590 and RR: 1.302, 95% CI: 1.051–1.614) than the lowest quartile. The highest quartile of SUA was related to the development of obesity (RR: 1.792, 95% CI: 1.028–3.121) (Table 2).

## Higher SUA related to worse renal outcomes

A total of 47 (0.4%) participants had 30% decline in eGFR during the follow up period. The highest quartile of SUA was significantly associated with 30% decline in eGFR in unadjusted and age- and sex- adjusted analysis. When risks were adjusted by multiple factors, the highest quartiles of SUA showed significant risks of 3.7 fold in 30% eGFR decline compared to corresponding risk of the lowest quartile (RR: 3.701, 95% CI: 1.504–9.108).

Incident CKD was developed in 101 (0.8%) participants. Models 1 and 2 showed significant associations of incident CKD with 4th quartiles of SUA. Model 3 showed that the highest SUA quartile had a 2.2 -fold (95% CI: 1.182–4.177, P = 0.013) increase in risk for incident CKD compared to the lowest quartile (Table 3).

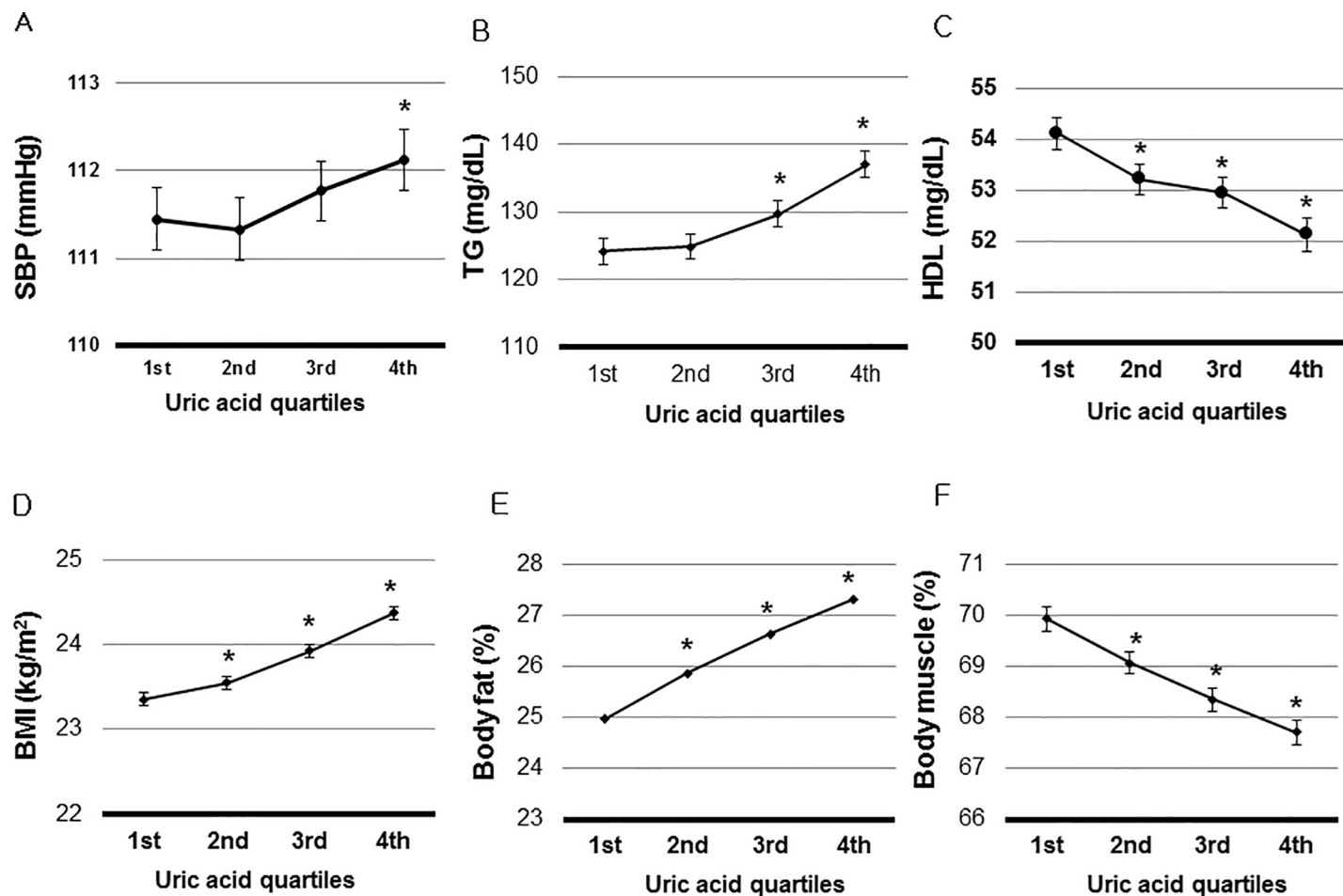

**Fig 2. Association of Serum Uric Acid (SUA) with metabolic disorders.** A. The highest quartile had significantly higher of systolic blood pressure (SBP) than the lowest quartile (P = 0.003). B. The 3rd and the highest quartiles had significantly higher triglyceride (TG) than the lowest quartile (P< 0.001). C. The 2nd, 3rd, and the highest quartiles had significantly higher high density lipoprotein (HDL) than the lowest quartile (P< 0.001). D. The 2nd, 3rd, and the highest quartiles had significantly higher BMI than the lowest quartile (P ≤0.024). E. The 2nd, 3rd, and the highest quartiles had significantly higher body fat percent than the lowest quartile (P ≤0.009). Body fat percent was measured in 3,682 participants. F. FBS was not associated with SUA (P >0.05). *P < 0.05 vs. the lowest uric acid quartile. SBP, TG, HDL, BMI, body fat percent, and FBS were adjusted by age, hemoglobin, white blood cell count, estimated glomerular filtration rate, serum albumin, aspartate aminotransferase, alanine aminotransferase, total cholesterol, and low density lipoprotein. Adjusted means and 95% confidence intervals (95% CIs) of SBP, TG, HDL, BMI, body fat percent and FBS were calculated using analysis of covariance to adjust independent factors related to SUA.

**Table 2. Risks for the development of hypertension, diabetes, and obesity according to the quartiles of baseline serum uric acid level.**

| | HTN | | | DM | | | Obesity | | |
|---|---|---|---|---|---|---|---|---|---|
| | RR | 95% CI | *P* | RR | 95% CI | *P* | RR | 95% CI | *P* |
| 1st | reference | | | reference | | | reference | | |
| 2nd | 0.987 | 0.788–1.234 | 0.906 | 1.165 | 0.832–1.631 | 0.373 | 0.801 | 0.403–1.591 | 0.526 |
| 3rd | 1.285 | 1.038–1.590 | 0.022 | 1.805 | 1.305–2.498 | <0.001 | 1.188 | 0.652–2.165 | 0.575 |
| 4th | 1.302 | 1.051–1.614 | 0.016 | 1.767 | 1.279–2.442 | 0.001 | 1.792 | 1.028–3.121 | 0.039 |

Abbreviations: hypertension (HTN), diabetes mellitus (DM)/

Relative risks (RR) and 95% confidence intervals (95% CIs) of development hypertension, diabetes and obesity were calculated using cox regression analysis.

Risks were adjusted by age, sex, systolic blood pressure, body mass index, estimated glomerular filtration rate, hemoglobin, white blood cell, low density lipoprotein, aspartate aminotransferase, alanine aminotransferase, fasting blood sugar, serum albumin, total cholesterol and malignancy.

**Table 3. Risks for the incidence of GFR decline and CKD development according to the quartiles of baseline serum uric acid level.**

| | Decline in eGFR of 30% | | | | | | | | |
|---|---|---|---|---|---|---|---|---|---|
| | Model 1 | | | Model 2 | | | Model 3 | | |
| | RR | 95% CI | *P* | RR | 95% CI | *P* | RR | 95% CI | *P* |
| 1st | reference | | | reference | | | reference | | |
| 2nd | 0.518 | 0.151–1.770 | 0.294 | 0.515 | 0.151–1.763 | 0.291 | 0.603 | 0.173–2.098 | 0.427 |
| 3rd | 1.702 | 0.679–4.267 | 0.257 | 1.736 | 0.692–4.354 | 0.239 | 2.098 | 0.814–5.406 | 0.125 |
| 4th | 3.011 | 1.292–7.019 | 0.011 | 3.067 | 1.316–7.153 | 0.009 | 3.701 | 1.504–9.108 | 0.004 |
| | Development of CKD | | | | | | | | |
| | Model 1 | | | Model 2 | | | Model 3 | | |
| | RR | 95% CI | *P* | RR | 95% CI | *P* | RR | 95% CI | *P* |
| 1st | reference | | | reference | | | reference | | |
| 2nd | 0.975 | 0.458–2.076 | 0.947 | 0.931 | 0.436–1.986 | 0.853 | 0.732 | 0.340–1.575 | 0.425 |
| 3rd | 1.274 | 0.624–2.601 | 0.505 | 1.428 | 0.699–2.917 | 0.329 | 0.803 | 0.385–1.677 | 0.560 |
| 4th | 3.964 | 2.168–7.249 | <0.001 | 4.853 | 2.643–8.910 | <0.001 | 2.222 | 1.182–4.177 | 0.013 |

Abbreviations: estimated glomerular filtration rate (eGFR), chronic kidney disease (CKD).

Relative risks (RR) and 95% confidence intervals (95% CIs) of decline in eGFR, and incident CKD were calculated using cox regression analysis.

Model 1: Unadjusted.

Model 2: adjusted by age, sex.

Model 3: adjusted by age, sex, systolic blood pressure, body mass index, estimated glomerular filtration rate, hemoglobin, white blood cell, low density lipoprotein, aspartate aminotransferase, alanine aminotransferase, fasting blood sugar, serum albumin, total cholesterol and malignancy.

## Discussion

In this study, SUA was associated with BMI, BP, dyslipidemia, and development of DM, HTN and obesity. High SUA is associated with increased risk of the development of DM, although SUA was not associated with FBS cross-sectionally. SUA was associated with decline in renal function and development of CKD. Interestingly, the risk of decline in renal function according to SUA was significant in participants without DM, HTN, and obesity.

The hypothesis that uric acid is a possible factor for renal disease is not new. Epidemiologic studies demostrated elevated serum uric acid as a risk factor for the development of acute and chronic kidney disease, hypertension, and diabetes [20]. A meta-analysis of 15 cohort studies showed that the relative risk of CKD was 1.22 per 1 mg/dL serum uric level increment [21]. SUA is an independent risk factor of cardiovascular disease and mortality [22, 23]. Animal experiments have reported that increases in serum uric acid by inhibiting urate oxidase can increase renal renin and decrease plasma nitrate, resulting in vasoconstriction and hypertension [24, 25]. Early hypertension is reversible by lowering uric acid. However, prolonged hyperuricemia can lead to activation of inflammatory mediators and grow factors, vascular smooth muscle cell proliferation, and vascular wall thickening. Reduction of SUA is insufficient to control high blood pressure in a prolonged state of hyperuricemia [5, 25, 26].

Accordingly, we hypothesized that SUA might be modifiable risk factor of renal outcome in participants without chronic vascular damage. Therefore, the association of SUA with renal outcome in participants with intact renal function was analyzed and significant risks were observed in participants with higher baseline SUA. A recent randomized trial did not demonstrate the improvement of renal function by urate lowering therapy in stage 3 CKD with asymptomatic hyperuricemia [13]. However, subgroup analysis showed significant benefit of uric acid lowering therapy in patients without proteinuria and with higher renal function [13]. An observational study in 3,885 individuals with CKD stages 2–4 also showed that SUA was an

independent risk factor for kidney failure only in earlier stages of CKD, but not associated with kidney failure in late stages of CKD [3]. A previous cohort study demonstrated that SUA is related to the new development of hypertension, but not to the incidence of CKD [27]. In this study, SUA is no longer an independent risk factor to predict the incidence of CKD by multivariate analysis and SUA is affected by the confounding effect of eGFR.

However, whether SUA is an independent risk factor for renal function decline remains unclear because metabolic profile at baseline is significant worse in participants with higher SUA. Hyperuricemia is related to obesity and insulin resistance [15, 16]. Increase intake of fructose and purine diet can cause hyperuricemia [14]. Plasma uric acid is elevated in obese mice due to production and secretion of uric acid in adipose tissue by increased production of xanthine oxidoreductase [28]. In addition, higher levels of circulating insulin can inhibit the excretion of uric acid through the kidney [29, 30]. To eliminate the effect of metabolic factors, we investigated the effect of SUA on renal outcome in participants without DM, HTN, and obesity. These associations of SUA with renal outcome was significant in such subjects without metabolic abnormalities. SUA is associated with systemic inflammatory markers in children with asymptomatic hyperuricemia [31]. Hyperuricemia causes increased of monocyte chemoattractant protein-1 and primes monocyte trafficking in gout patients [32]. In this study, we observed SUA was independently associated with inflammatory markers such as WBC and CRP. SUA is associated with increased arterial stiffness and decreased renal function in normotensive individuals [33]. Irrespective of metabolic abnormalities, SUA might contribute to the development of CKD by increasing inflammation and vascular stiffness.

The relationship of FBS with SUA remains controversial. Many studies have reported positive relationships between serum glucose and SUA [14–16, 18]. One study on 11,183 with normal glucose tolerance has demonstrated that a U-shape relationship between SUA and FBS [34]. By contrast, SUA is inversely related to FBS in studies conducted on healthy population [35, 36]. The third National Health and Nutrition Examination Survey showed that SUA levels were negatively associated with type 2 DM [35]. The relationship between SUA and FBS is currently unclear. Approximately 70% of uric acid is excreted by the kidney. Among freely filtered uric acid through glomerulus, 90% is reabsorbed through proximal tubule [37]. Low SUA was observed in participants treated with sodium glucose cotransporter 2 inhibitors. Such low SUA may be due to increased excretion of uric acid in proximal renal tubule by increased glycosuria [38, 39]. In our study, cross-sectional analysis revealed that SUA was not associated with FBS after the adjustment. Uricosuric effect in participants with high amount of glycosuria could affect the cross-sectional relationship between SUA and FBS. However, and the incidence of DM was increased in participants with higher SUA in accordance with increased development of obesity after a long-term follow up.

This study has several limitations. First, we did not have data of medications. Therefore, the prevalence of HTN, DM, and dyslipidemia might have been underestimated. Second, this study was performed on a healthy population. The association of SUA with renal outcome could be different in CKD patients. Third, we included the participants who measured serum creatinine two times, and the defining CKD could be overestimated because there is a possibility that we included the participants who had a transient impairment of renal function at the follow up study. Finally, this study only investigated Koreans. However, data were analyzed cross-sectionally and longitudinally. In addition, we could investigate temporal relationship of SUA with metabolic abnormalities and renal outcomes. We also investigated the relationship of SUA with renal outcomes in participants without metabolic abnormalities.

In conclusion, high SUA is an independent risk factor for the development of diabetes, hypertension, and obesity in the healthy population. High SUA is associated with increased risk of CKD development and eGFR decline in participants with intact renal function.

## Acknowledgments

We would like to thank Hyeon Jin Min and Bong Gyun Sun for the contribution of this work.

## Author Contributions

**Conceptualization:** Se Won Oh.

**Data curation:** Young-Bin Son, Se Won Oh.

**Formal analysis:** Young-Bin Son, Se Won Oh.

**Funding acquisition:** Se Won Oh.

**Investigation:** Se Won Oh.

**Methodology:** Se Won Oh.

**Project administration:** Se Won Oh.

**Resources:** Se Won Oh.

**Software:** Se Won Oh.

**Supervision:** Young-Bin Son, Ji Hyun Yang, Myung-Gyu Kim, Sang Kyung Jo, Won Yong Cho, Se Won Oh.

**Validation:** Myung-Gyu Kim, Sang Kyung Jo, Won Yong Cho, Se Won Oh.

**Visualization:** Se Won Oh.

**Writing – original draft:** Young-Bin Son, Se Won Oh.

**Writing – review & editing:** Young-Bin Son, Se Won Oh.

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
