## [Decision Letter · Decision Letter 0]

5 Aug 2020

PONE-D-20-13895

Serum uric acid and risk of chronic kidney disease in normotensive, normoglycemic, and non-obese individuals

PLOS ONE

Dear Dr.  Se Won Oh 

Thank you for submitting your manuscript to PLOS ONE. After careful consideration, we feel that it has merit but does not fully meet PLOS ONE’s publication criteria as it currently stands. Therefore, we invite you to submit a revised version of the manuscript that addresses the points raised during the review process.

It would be important for the authors to try to revise the comments made by this editor and regarding the statistics observed by one of the referees with whom I agree that it will certainly improve the study.

The study was well conducted but the authors need to analyze some points that I consider to be important in the manuscript, and that, if modified, will contribute to improving the text. Suggestions:

- the summary would need to be reformulated as it does not make it clear that it isis a follow-up study;

- in methods it would be necessary to describe the power of the sample used in the study and also it is important to describe in the results all the variables used;

- what were the analyzes carried out in order to verify if the ANOVA test is suitable for the study, and it would also be important to consider if the results obtained do not depend only on the high number of individuals collected in the study, therefore, they are really representative of the studied population, and,  

- when looking for risk calculation I believe it is better to use logistic regression analysis.

I hope that the authors will consider the suggestions made by the referees and this editor revising the manuscript..

We look forward to receiving your revised manuscript.

Kind regards,

Dulce Elena Casarini, PhD, FAHA

Academic Editor

PLOS ONE

Additional Editor Comments:

The study was well conducted but the authors need to analyze some points that I consider to be important in the manuscript, and that, if modified, will contribute to improving the text.

Suggestions:

- the summary would need to be reformulated as it does not make it clear that it isis a follow-up study;

- in methods it would be necessary to describe the power of the sample used in the study and also it is important to describe in the results all the variables used;

- what were the analyzes carried out in order to verify if the ANOVA test is suitable for the study, and it would also be important to consider if the results obtained do not depend only on the high number of individuals collected in the study, therefore, they are really representative of the studied population, and,

- when looking for risk calculation I believe it is better to use logistic regression analysis.

Journal Requirements:

2. Please address the following:

- Please refer to any post-hoc corrections to correct for multiple comparisons during your statistical analyses. If these were not performed please justify the reasons. Please refer to our statistical reporting guidelines for assistance (https://journals.plos.org/plosone/s/submission-guidelines.#loc-statistical-reporting).

- Please provide the dates upon which the patient data was accessed.

Reviewers' comments:

Reviewer's Responses to Questions

**Comments to the Author**

1. Is the manuscript technically sound, and do the data support the conclusions?

Reviewer #1: Yes

Reviewer #2: Partly

2. Has the statistical analysis been performed appropriately and rigorously? 

Reviewer #1: Yes

Reviewer #2: No

3. Have the authors made all data underlying the findings in their manuscript fully available?

Reviewer #1: Yes

Reviewer #2: No

4. Is the manuscript presented in an intelligible fashion and written in standard English?

Reviewer #1: Yes

Reviewer #2: Yes

5. Review Comments to the Author

Reviewer #1: Dear authors

Congratulations

You -the authors of the current research studied the association between serum uric acid levels and declining of renal function in individuals without comorbidities from Korea. You observed that SUA is an independent risk factor for declining eGFR, even in individuals with a normal range of renal function and without metabolic disorder. Despite being a retrospective study, you show interesting data showing the relationship that was the objective of the current research.

But, I need to make some small comments, such as...

In page 4

materials and methods

-It is necessary a figure with flowchart showing enrolment and exclusion criterias of individuals in this study.

In page 7; the results part

Higher SUA related to worse renal outcomes

I guess is better to describe:

-When risks were adjusted by multiple factors, the highest quartiles of SUA showed significant risks of 3.7 fold in eGFR decline compared to corresponding risk of the lowest quartile (RR: 3.701, 95% CI: 1.504-9.108).

-It is correct to assume that "...highest SUA quartile had a 2.222-fold (95% CI: 1.182-4.177, P =0.013) increase in risk for incident CKD...".

I see that is better "...highest quartile had a 2.2 fold (95% CI: 1.182-4.177) increase in risk for incident chronic kidneyd isease (CKD)".

Reviewer #2: This is an interesting paper and appears to have approached an important question regarding whether high levels of uric acid can be associated with loss in the renal function in adult population. However, there are several important issues that require clarification.

1) The title does not reflect the findings of the manuscript. It is not clear that this is a follow-up study. After the follow-up, there are already hypertensive and metabolically unhealthy individuals. Therefore, the title is inappropriate.

2) Please, do not use abbreviations in the summary section

3) The summary is not adequate. It is not clear in the aim section that this is a follow-up study.

4) Describe how the patients were recruited so that the reader can determine how representative this sample is of the population. For example, were these consecutive patients presenting to the Center?

5) Blood pressure data cannot be represented in decimals, please correct.

6) In the methodology section, it is important to describe all the variables present in the results.

7) Were normality tests performed?

8) How was the body fat percentage obtained?

9) What diagnostics were done to check that ANOVA fit and that results were not dependent on only the high number of individuals?

10) Wouldn't it be more informative to examine the baseline data by correlation analysis.

11) In Table 1, it appears that many variables do not show a linear trend across the quartiles of uric acid (e.g. Hb, AST, ALT, ALP, bilirubin, protein, albumin, glucose). This contradicts the claim that there is a linear trend (P < 0.001).

12) It would be more appropriate to perform logistic regression for the risk calculation

13) It would be appropriate to perform Poisson Regression

14) Results (Figures): The graphic quality is very poor.

15) The discussion section is very simple and could be better detailed.

16) The conclusion is confusing and speculative.

6. PLOS authors have the option to publish the peer review history of their article (what does this mean?). If published, this will include your full peer review and any attached files.

Reviewer #1: No

Reviewer #2: No

---

## [Author Response · Author response to Decision Letter 0]

18 Oct 2020

Dear reviewers

Thank you for your thoughtful comments regarding our manuscript. We took your comments seriously and revised our manuscript again. 

Reviewer #1: Dear authors

Congratulations

You -the authors of the current research studied the association between serum uric acid levels and declining of renal function in individuals without comorbidities from Korea. You observed that SUA is an independent risk factor for declining eGFR, even in individuals with a normal range of renal function and without metabolic disorder. Despite being a retrospective study, you show interesting data showing the relationship that was the objective of the current research.

But, I need to make some small comments, such as...

In page 4

materials and methods

-It is necessary a figure with flowchart showing enrolment and exclusion criterias of individuals in this study.

Thank you for your correction. I added flowchart of study population. 

In page 7; the results part

Higher SUA related to worse renal outcomes

I guess is better to describe:

-When risks were adjusted by multiple factors, the highest quartiles of SUA showed significant risks of 3.7 fold in eGFR decline compared to corresponding risk of the lowest quartile (RR: 3.701, 95% CI: 1.504-9.108).

-It is correct to assume that "...highest SUA quartile had a 2.222-fold (95% CI: 1.182-4.177, P =0.013) increase in risk for incident CKD...".

I see that is better "...highest quartile had a 2.2 fold (95% CI: 1.182-4.177) increase in risk for incident chronic kidneyd isease (CKD)".

Thank you for your correction. I changed the manuscript as your suggestions.

Reviewer #2: This is an interesting paper and appears to have approached an important question regarding whether high levels of uric acid can be associated with loss in the renal function in adult population. However, there are several important issues that require clarification.

1) The title does not reflect the findings of the manuscript. It is not clear that this is a follow-up study. After the follow-up, there are already hypertensive and metabolically unhealthy individuals. Therefore, the title is inappropriate.

Thank you for your comments. I changed the title as “The effect of baseline serum uric acid on chronic kidney disease in normotensive, normoglycemic, and non-obese individuals: A health checkup cohort study.”

2) Please, do not use abbreviations in the summary section

Thank you for your correction. I added full terminology before the use of abbreviations in the abstract section.

3) The summary is not adequate. It is not clear in the aim section that this is a follow-up study.

Thank you for your comments. I added the comments regarding follow-up study in the abstract section. 

4) Describe how the patients were recruited so that the reader can determine how representative this sample is of the population. For example, were these consecutive patients presenting to the Center?

Korea University Anam Hospital is a tertiary hospital and has a health checkup center. Participants voluntarily came to the hospital and performed general health checkup. Among them, approximately 30.8% of participants retested health checkup voluntarily. 

5) Blood pressure data cannot be represented in decimals, please correct.

I corrected blood pressure data in table 1. 

6) In the methodology section, it is important to describe all the variables present in the results.

Thank you for your comments. I added about the description of all the variables in the methodology section. 

7) Were normality tests performed?

We performed normality tests in all variables in this study. As previous studies documented (Am J Med. 1965; 39: 242-51, J Clin Hypertens (Greenwich). 2017 Jan;19(1):45-50), serum uric acid was not normally distributed and skewed to the high value end of the scale ( Kolmogorov-Smirnov test, P >0.05).

 Average= 5.21; standard deviation= 1.375; N= 13,133

7) How was the body fat percentage obtained?

We measured total body fat mass and lean body mass using bioelectrical impedance analysis (InBody770, InBody Co, Ltd., Seoul, Korea). And fat percentage was calculated based on body weight (fat mass/ body weight *100). 

8) What diagnostics were done to check that ANOVA fit and that results were not dependent on only the high number of individuals?

Only the high values of serum uric acid (the highest quartile of serum uric acid) showed a significant risk increase for development of chronic kidney disease and eGFR decline, and other groups of serum uric acid (1st, 2nd 3rd quartile of serum uric acid) was not associated with development of chronic kidney disease and eGFR decline. We analyzed these results using cox regression analysis. ANOVA was not used.

9) Wouldn't it be more informative to examine the baseline data by correlation analysis.

 I re-examined baseline data by correlation analysis between serum uric acid with many variables.

Serum uric acid was positively associated with systolic blood pressure (P<0.001, r=0.289), diastolic blood pressure (P<0.001, r=0.304), body mass index (P<0.001, r=0.363), white blood cell count (P<0.001, r=0.218), hemoglobin (P<0.001, r=0.565), serum protein (P<0.001, r=0.125), albumin (P<0.001, r=0.221), AST (P<0.001, r=0.136), ALT (P<0.001, r=0.230), ALP (P<0.001, r=0.173), bilirubin (P<0.001, r=0.236), total cholesterol (P<0.001, r=0.147), fasting blood glucose (P<0.001, r=0.168), triglyceride (P<0.001, r=0.331), LDL (P<0.001, r=0.190), and C-reactive protein (P<0.001, r=0.067). In addition, serum uric acid was negatively correlated with eGFR (P<0.001, r=-0.283) and HDL (P<0.001, r=-0.299). 

10) In Table 1, it appears that many variables do not show a linear trend across the quartiles of uric acid (e.g. Hb, AST, ALT, ALP, bilirubin, protein, albumin, glucose). This contradicts the claim that there is a linear trend (P < 0.001).

In table 1, you can see the mean values such as Hb, AST, ALT, ALP, bilirubin, protein, albumin and glucose are increased according to the increase of uric acid quartiles. There is a linear trend between the quartiles of uric acid and following variables: age, body mass index, systolic blood pressure, diastolic blood pressure, eGFR, hemoglobin, white blood cell count, AST, ALT, ALP, bilirubin, protein, albumin, glucose, cholesterol, TG, HDL, LDL and CRP (P for trend ≤ 0.001). 

11) It would be more appropriate to perform logistic regression for the risk calculation.

In table 2 and 3, we used cox regression analysis because logistic analysis cannot adjust the follow up time period. We wanted to evaluate the risk for the development of HTN, DM, obesity, and renal outcomes during follow up period. In figure 1, we used analysis of covariance to investigate the association of baseline serum uric acid with SBP, TG, HDL, BMI, body fat percent, and FBS because dependent variables were all continuous variables. Dependent variable of logistic analysis should be categorical variable. 

12) It would be appropriate to perform Poisson Regression.

Unfortunately, I cannot understand fully about your comment. The dependent variables such as the development of hypertension, diabetes, obesity, eGFR decline or chronic kidney disease were “dichotomous” scale (none or presence). The events could occur only one time during follow up period and participants got a disease state after the events. However, the poisson regression may be more appropriate when the dependent variables is “count scale”: more than zero or greater. 

13) Results (Figures): The graphic quality is very poor.

Thank you for your comment. I corrected the graphic.

14) The discussion section is very simple and could be better detailed.

Thank you for your comments. I added following sentences in the discussion section.

15) The conclusion is confusing and speculative.

Thank you for your comments. I corrected the conclusions as follows. 

: In conclusion, high SUA is an independent risk factor for the development of diabetes, hypertension, and obesity in the healthy population. High SUA is associated with increased risk of CKD development and eGFR decline in participants with intact renal function.

Sincerely,

Se Won Oh

Associate professor 

Division of Nephrology, Department of Internal Medicine, Korea University Anam Hospital, Korea University College of Medicine

73, Inchon-ro, Seongbuk-gu, Seoul, 02841, Republic of Korea; Tel: 82-2-920-6260; e-Mail: hisy81@hanmail.net

---

## [Decision Letter · Decision Letter 1]

3 Dec 2020

The effect of baseline serum uric acid on chronic kidney disease in normotensive, normoglycemic, and non-obese individuals: A health checkup cohort study

PONE-D-20-13895R1

Dear Dr. Oh,

We’re pleased to inform you that your manuscript has been judged scientifically suitable for publication and will be formally accepted for publication once it meets all outstanding technical requirements.

Kind regards,

Cheng Hu

Academic Editor

PLOS ONE

Additional Editor Comments (optional):

Reviewers' comments:

Reviewer's Responses to Questions

**Comments to the Author**

1. If the authors have adequately addressed your comments raised in a previous round of review and you feel that this manuscript is now acceptable for publication, you may indicate that here to bypass the “Comments to the Author” section, enter your conflict of interest statement in the “Confidential to Editor” section, and submit your "Accept" recommendation.

Reviewer #1: All comments have been addressed

Reviewer #2: All comments have been addressed

2. Is the manuscript technically sound, and do the data support the conclusions?

Reviewer #1: Yes

Reviewer #2: Yes

3. Has the statistical analysis been performed appropriately and rigorously? 

Reviewer #1: Yes

Reviewer #2: Yes

4. Have the authors made all data underlying the findings in their manuscript fully available?

Reviewer #1: Yes

Reviewer #2: Yes

5. Is the manuscript presented in an intelligible fashion and written in standard English?

Reviewer #1: Yes

Reviewer #2: Yes

6. Review Comments to the Author

Reviewer #1: The authors answered all doubts and questions made to them about the current manuscript. The authors aligned the title with the objectives and the conclusion. They made a new chart. They managed to mitigate the statistical deficiency. They managed to improve the discussion of the manuscript. In my opinion, the manuscript is now ready to be published.

Reviewer #2: (No Response)

7. PLOS authors have the option to publish the peer review history of their article (what does this mean?). If published, this will include your full peer review and any attached files.

Reviewer #1: No

Reviewer #2: No

---

## [Editor Report · Acceptance letter]

13 Jan 2021

PONE-D-20-13895R1 

The effect of baseline serum uric acid on chronic kidney disease in normotensive, normoglycemic, and non-obese individuals: A health checkup cohort study 

Dear Dr. Oh:

I'm pleased to inform you that your manuscript has been deemed suitable for publication in PLOS ONE. Congratulations! Your manuscript is now with our production department. 

Kind regards, 

on behalf of

Dr. Cheng Hu 

Academic Editor

PLOS ONE